REGISTERED REPORT PROTOCOL

# Incidence of lung cancer and mortality among civil construction industry workers: A protocol for a systematic review and meta-analysis

Rita Stella Maria Cahuana Pinto[1], Alana Castro Panzenhagen[2], Luis Felipe Silva Oliveira[1,3], José Claudio Fonseca Moreira[2,4], Carlos Eduardo Schnorr[5]*

1 Departamento de Civil y Ambiental, Universidad de la Costa, Barranquilla, Atlántico, Colombia, 2 Programa de Pós-graduação em Ciências Biológicas: Bioquímica, Universidade Federal do Rio Grande do Sul, Porto Alegre, Rio Grande do Sul, Brasil, 3 Universidad de Lima, avenida Javier Prado Este, Santiago de Sucro, Perú, 4 Programa de Pós-graduação em Biologia Celular e Molecular, Universidade Federal do Rio Grande do Sul, Porto Alegre, Rio Grande do Sul, Brasil, 5 Departamento de Ciencias Naturales y Exactas, Universidad de la Costa, Barranquilla, Atlantico, Colombia

* cschnorr@cuc.edu.co

## Abstract

### Background

The construction sector is one of the most stable growth industries in the world. However, many studies have suggested an association between occupational exposure in civil construction and lung cancer risk. Thus, this study aims to assess lung cancer risk in civil construction workers occupationally exposed to physical and chemical agents through a systematic review and meta-analysis.

### Methods/design

Studies will be identified by searching PUBMED, Embase, SCOPUS, WEB OF SCIENCE and the reference list of included articles. Eligible study designs will be cohort, cross-sectional, and case-control studies that report occupational exposure to physical or chemical agents and lung cancer risk through mortality or incidence outcomes. A meta-analysis will be used to combine odds ratios (ORs) from case-control studies and relative risks (RR) from cohort studies. Two reviewers will independently screen articles, extract data, and assess scientific quality using standardized forms and ROBINS-E tool if available. Otherwise, the New-Castle Ottawa rating scale will be used. Any of those will also be used in combination with the GRADE approach for quality of evidence. Overall risk estimates and their corresponding 95% confidence intervals (CIs) will be obtained using the random-effects model meta-analysis. This systematic review and meta-analysis will be conducted following the Meta-analysis of Observational Studies in Epidemiology (MOOSE) guidelines. Results will be reported according to the Preferred Reporting Items for Systematic Reviews and Meta-Analyses (PRISMA) statement.

### Discussion

This review will identify and synthesize studies investigating the association between occupational exposure in the construction industry and lung cancer. The findings will help

This is a Registered Report and may have an associated publication; please check the article page on the journal site for any related articles.

**Data Availability Statement:** All relevant data from this study are publicly available here: https://osf.io/bgz2a/ (DOI: 10.17605/OSF.IO/BGZ2A).

**Funding:** The author(s) received no specific funding for this work.

**Competing interests:** The authors have declared that no competing interests exist.

governmental entities and researchers with evidence-based decision-making because they will integrate and validate the evidence on construction workers' health effects due to occupational exposure.

## Systematic review registration

PROSPERO CRD42020164209

## Introduction

The construction sector is one of the most stable growth industries globally and represents up to 7% of the global workforce and a 15% share of the world's Gross Domestic Product [1]. However, construction activity is also one of the leading agents in contributing to environmental pollution despite its importance for a country's economic growth. Indeed, in 2018 buildings and construction sectors alone were responsible for 36% of the energy consumption and 39% of carbon dioxide ($CO_2$) emissions worldwide [2]. Additionally, occupational exposure in the construction industry puts the health of construction workers at risk as they may be exposed to physical and chemical agents. Indeed, a construction worker might be potentially exposed to more than 70 different substances, including materials related to their specific trade and of other businesses in their shared workplaces [3]. The list includes many chemicals known to have adverse health effects, such as natural and artificial mineral fibers, cement, quartz, miscellaneous powders, diesel exhaust, paints, and solvents [4, 5].

Furthermore, some exposure to several substances commonly found in the construction industry environment may also be associated with an increased risk of lung cancer [6]. For instance, asbestos, wood dust, crystalline silica, chromium (VI), lead and nickel compounds, benzene, and polycyclic aromatic hydrocarbons (PAH), are indeed classified by the International Agency of Research on Cancer as known carcinogens [7]. Additionally, several primary studies have investigated the risk of lung cancer in civil construction workers, including cohort and case-control studies. Many of these have reported consistent evidence for increased mortality. Still, others have shown no sufficient evidence for increased risk [8–16].

Moreover, specific trades in the construction industry have also been investigated. One study found that bricklayers had an increased risk of lung cancer within the SYNERGY database of case-control studies [17, 18]. Another study suggested an increased risk of lung cancer in Italian bricklayers from exposure to various carcinogens, especially crystalline silica [17, 18]. Finally, another research work found an increased risk of lung cancer for all construction occupations, except managers, engineers, and supervisors [19].

Furthermore, some systematic reviews and meta-analyses have been performed to evaluate better the association between lung cancer and occupational exposure in the construction industry. Some systematic reviews have investigated the evidence regarding specific chemical exposures and lung cancer risks, and others have included both construction and non-construction workers [20–22]. Nevertheless, none of the previous systematic reviews and metanalysis have considered the civil construction industry's specific characteristics and workers. It seems pertinent to integrate the information on lung cancer risk associated with physical and chemical agents commonly in contact with workers of different trades and occupations. Thus, we will present a systematic review and meta-analysis protocol to synthesize and evaluate the association between occupational exposure to physical, chemical agents and lung cancer risk

in construction workers. We consider this synthesis as paramount to obtain more accurate and valid estimates of lung cancer risk incidence and its effect magnitude.

## Review objectives

Our systematic review and meta-analysis's main objective will be to summarize the evidence on how working in the civil construction industry impacts lung cancer risk for workers. The questions we intend to answer are:

Are construction industry workers who have been exposed to physical and chemical agents at increased risk of lung cancer through mortality and incidence outcomes? Furthermore, if enough studies are available, quantify it through a meta-analytic approach. Are the incidence or mortality different among the workers within the many specialized and unspecialized trades in the civil construction industry? What physical or chemical agents are present in the construction industry environment associated with lung cancer risk in construction workers?

# Methods/design

## Protocol

The systematic review and meta-analysis will be conducted following the Meta-analysis of Observational Studies in Epidemiology (MOOSE) guidelines [23] and reported following the Preferred Reporting Items for Systematic Reviews and Meta-Analyses (PRISMA) statement [24]. This protocol has been registered within the International Prospective Register of Systematic Reviews (PROSPERO) database (registration number: CRD42020164209).

## Ethics

This review does not require ethical approval as the review is entirely based on the published data of the ethically approved primary studies.

## Criteria for selecting studies for this review

The following criteria will be used to identify studies to be included in this review.

**Inclusion and exclusion criteria.** Studies will be included if they meet the following criteria:

The study reports enough data for calculating odds ratios (OR) or relative risks (RR) and 95% confidence intervals (CI). Intervals can also be imputed when they are not available. In the case of duplicated or shared data from the same population, we will include the study with the largest sample size.

The study has a control group with limited exposure to airborne chemical pollutants at construction sites (white-collar, managers, engineers, supervisors, office workers).

Studies not meeting the inclusion criteria described above will be excluded. Further, animal studies, studies that do not assess the risk for lung cancer in humans, and studies without a proper control group will also be excluded.

**Participants/population.** We will include studies that investigate effects on participants (both male and female) who are construction workers, with unequivocal evidence of occupational exposure to physical and chemical agents, such as, but not limited to: carpenters, floor layers, bricklayers, painters, electricians, plumbers, welders, scaffolders, roofers, masons, sheet metal workers, rebar workers, construction laborers, machine operators, drywall installers, and insulators. No lower or upper limits will be set to the age of the participants included in the original studies.

**Exposure.** This review will include studies that report occupational exposure at a construction site, including short-term (up to seven days) and cumulative (for more than seven days) studies that evaluate occupational exposure with a risk matrix through operational inspection, questionnaires, or specialized measuring equipment.

**Comparator(s)/control.** The control group will be white-collar managers, engineers, office workers, and supervisors in the construction industry.

**Outcome.** The primary outcome will be lung cancer risk (through mortality and incidence outcomes) based on clinically confirmed diagnosis (death certificates, cancer registry, other national recording systems, or hospital records). The effects measured will include the odds ratios (OR), relative risks (RR), and 95% confidence intervals (CI). Other summary statistics data will be sought whenever OR or RR are not available.

**Study design.** Eligible studies will be comparative observational studies that report occupational exposure to physical and chemical agents in the civil construction industry and the outcome of interest (lung cancer risk through mortality and incidence outcomes). Cohort, comparative cross-sectional, and case-control studies will be included.

We will exclude studies in which lung cancer is not reported as an outcome of interest. Observational studies not presenting study-specific data (e.g., relative risks, 95% confidence intervals, numbers of cases/population, observed and expected cases) or sufficient data for an outcome measure to be calculated will also be excluded.

## Search strategy

We will identify potentially relevant studies by searching multiple electronic databases and websites such as PubMed, Embase, Scopus, and Web of Science. Keywords related to construction workers, occupational exposure, physical and chemical agents, and lung cancer will be used. The search strategy will be adapted for each database. The search terms will be: ("construction workers" OR "bricklayers" OR "painters" OR "carpenters" OR "floor layers" OR "electricians" OR "plumbers" OR "scaffolders" OR "roofers" OR "masons" OR "sheet metal workers" OR "rebar workers" OR "machine operators" OR "drywall installers" OR "insulators" OR "white collar workers" OR "supervisor" OR "managers" OR "engineer") AND ("construction industry" OR "construction material" OR "construction site" OR "occupational exposure" OR "construction emission" OR "physical agent" OR "chemical agent" OR "construction dust" OR "construction activity" OR "inorganic dust construction" OR "particulate matter" OR "silica" OR "asbestos" OR "cement" OR "release agents" OR "concrete" OR "polyurethane" OR "resins" OR "paints" OR "varnishes" OR "solvents") AND ("lung cancer" OR "lung function decline" OR neoplasm OR "lung diseases" OR carcinoma OR mesothelioma OR adenocarcinoma OR "pulmonary neoplasm" OR "pulmonary cancer" OR "small cell lung cancer").

Additional studies will be identified from the reference list of included articles and relevant reviews. No date restrictions will be imposed. There will be a restriction by the language of publication, only including English- and Spanish- written reports. Searches will be conducted from inception of the databases and updated to at least six months before submitting the final manuscript.

## Study selection

Study selection will be conducted in three stages. First, the resulting citations will be stored in the Rayyan platform (https://rayyan.qcri.org/) and will be screened for duplicates. Second, all titles and abstracts will be independently screened by two reviewers against the inclusion/ exclusion criteria to identify potentially relevant studies. Studies that do not meet specific inclusion/exclusion criteria will be rejected at this stage, and the reason for rejection will be

recorded. Third, the full-text articles of all remaining studies will be obtained and independently assessed for inclusion by two reviewers. Disagreements between the two reviewers will be resolved by discussion, with the involvement of a third reviewer. Multiple reports of the same study will be counted only once; the record containing the most outstanding amount of information (for example, the largest sample size or the most prolonged follow-up period) will be retained. A PRISMA flow diagram showing details of studies included and excluded at each stage of selection will be produced.

## Data extraction

Data from all included studies will be extracted independently by two reviewers using a standardized data extraction sheet piloted on a sample of five studies and then modified, if necessary, before full data extraction begins. Any disagreements between the two reviewers at any stage will be referred to a third reviewer for the final decision. For case-control studies, we will extract the sample size of the case and control groups, the OR, and the 95% CI for lung cancer from each included study. Likewise, we will extract the number of observed deaths or cases for cohort studies, the number of expected fatalities or cases, the RR, and the 95% confidence interval (CI) for lung cancer risk. We will perform the analysis using RevMan (version 5.4.1) software. For all included studies, the following data will also be extracted where available: first author, publication year, geographic area, study type, industry type, occupation, pollutant type, total number of cases (sample size) and controls, sex, age, smoking history, measurement of exposure and assessment method (medical diagnosis, spirometry or questionnaires), the definition of occupational exposure, the period of employment/exposure, case definition, type of risk, and classification of the outcome.

## Quality assessment

Two reviewers will independently screen articles, extract data, and assess scientific quality using standardized forms and a published quality assessment tool (ROBINS-E tool will be used in case it is already available at https://www.bristol.ac.uk/population-health-sciences/centres/cresyda/barr/riskofbias/robins-e/). Otherwise, the New-Castle Ottawa rating scale will be used [25, 26]. This assessment scale consists of eight items categorized into three broad perspectives: selecting the study groups, comparability of the groups, and exposures or outcomes of interest. For cohort studies, a modified version of the NOS will be used. This modified NOS was developed for assessing the quality of occupational cohort studies and included five quality components: representativeness of the exposed cohorts, exposure assessment/reporting, comparability of the exposed and non-exposed cohorts, assessment of outcome, and adequacy of follow-up [26]. Any of those methods will also be used together with the GRADE approach for quality of evidence through GRADEpro (https://gdt.gradepro.org/app/) [27].

## Data synthesis

A meta-analysis will be performed when at least three studies investigating the same outcome are available. A random-effects model [28] and OR or RR effect measures will be used. A systematic narrative review will be conducted through qualitative analysis in case of lack of data or very high heterogeneity levels (> 85%). Heterogeneity among studies will be investigated using the $I^2$ statistic and Cochran's Q test [29]. Thresholds for the $I^2$ statistics <25% will be taken to suggest low heterogeneity, <50% to suggest moderate heterogeneity, and >75% to suggest high heterogeneity, while the significance level for chi-squared will be set at P = 0.1. Publication bias will be investigated by visual inspection of Begg's funnel plots, a scatter plot of

the studies constructed from a meta-analysis [28]. We will also perform Egger's regression tests if ten or more studies are included in a meta-analysis [30].

The lung cancer risk in workers may be treated separately in the outcome analysis, considering the subgroup analyses or meta-regression. The heterogeneity and the number of studies available will be considered before the decision to conduct subgroup analysis. If possible, we will use subgroup and meta-regression analyses to explain any observed between-study heterogeneity if at least ten studies are included in a meta-analysis to allow statistical power for such investigations. The covariates considered will be the geographic region, sample size, follow-up period, type of exposure, type of industry, occupation, lung cancer type, the national prevalence of cigarette smoking, and national risk rate for lung cancer.

For sensitivity analyses, we will analyze critical outcomes classified by categories of the assessed study quality variables to ascertain whether there are any relations with quality and outcome. Additionally, we will assess the individual studies' influence on results by re-estimating the overall effect after omitting each study in turn (Jackknife method). We will also conduct a cumulative meta-analysis in order of publication year to find the starting point of risk estimate becoming statistically significant and clarify the variation tendency [28].

The meta-analysis will be performed using RevMan (version 5.4.1) software, while regression analyses and funnel plots will be generated with the R environment package *meta*.

## Discussion

A systematic review and metanalysis will help us to identify and synthesize the evidence of the association between exposure to physical, chemical agents and lung cancer in construction workers. The systematic review findings will also help us better understand the risk differences among workers of specific trades or occupations. Indeed, construction workers are an extensive and diverse group of workers and include specialized workers, such as painters, carpenters, plumbers, electricians, roofers, and many other trades and many unspecialized workers who support the former in their activities. This means that the exposure and risk might be stratified along those various group pf workers, in which some could be more at risk than others. We will also compare our results with previous systematic reviews about specific chemical exposures or reviews that have included construction and non-construction workers [20–22]. It will allow us to evaluate how the exposure in the construction industry context is similar or different from other industrial settings or other study populations.

Additionally, systematic reviews and meta-analyses are the principal methodologies to obtain a solid and truthful synthesis of scientific evidence since systematic reviews represent the highest hierarchical level of evidence [31] given the existent literature. Moreover, a systematic review on the subject will provide data on the methodology of the different studies and the strengths of the published literature, which can help the development of new experimental designs; identify the reasons for the discrepancies or contradictions between the results of the different investigations, encouraging the redesign of the studies to improve the existing research methods; and provide a more precise estimate of the effect magnitude of scientific evidence for the subject.

The results obtained from our study will: 1) provide consistent information about the affectation to the health of construction workers by occupational exposure to physical and chemical agents; and 2) be valuable to occupational health and public health policymakers to minimize the exposure of workers to carcinogenic agents. We hope that in summarizing and thoroughly evaluating primary studies' consistency and quality we will be contributing to better practice decision making as well as the implementation of evidence-based prophylactic measures to prevent lung cancer in specialized lines of work.

Nonetheless, our planned methods have some limitations. The principal limit is the quality and thoroughness of primary studies; for example, the methods used and exposure classification vary. It will also be essential to highlight the heterogeneity between the studies regarding the lung cancer risk result, with considerable differences in occupation, contaminating substances, type of lung cancer, duration of exposure, etc.

We plan to disseminate the results of this systematic review and metanalysis to researchers and research funders in scientific conferences and by publication in a peer-review journal relevant to the field and policymakers by the publication of print materials and personal communication.

## Supporting information

**S1 Checklist. PRISMA-P checklist.**
(DOC)

## Author Contributions

**Conceptualization:** Rita Stella Maria Cahuana Pinto, Carlos Eduardo Schnorr.

**Investigation:** Rita Stella Maria Cahuana Pinto, Alana Castro Panzenhagen, Carlos Eduardo Schnorr.

**Methodology:** Rita Stella Maria Cahuana Pinto, Alana Castro Panzenhagen, José Claudio Fonseca Moreira, Carlos Eduardo Schnorr.

**Project administration:** Rita Stella Maria Cahuana Pinto.

**Supervision:** Carlos Eduardo Schnorr.

**Validation:** Rita Stella Maria Cahuana Pinto, Alana Castro Panzenhagen, Luis Felipe Silva Oliveira, José Claudio Fonseca Moreira, Carlos Eduardo Schnorr.

**Writing – original draft:** Rita Stella Maria Cahuana Pinto.

**Writing – review & editing:** Alana Castro Panzenhagen, Luis Felipe Silva Oliveira, José Claudio Fonseca Moreira, Carlos Eduardo Schnorr.

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
