## [Decision Letter · Decision Letter 0]

9 Dec 2020

PONE-D-20-29581

Lung cancer risk in workers of the civil construction industry:  A protocol for a systematic review and meta-analysis

PLOS ONE

Dear Dr. Schnorr,

Thank you for submitting your manuscript to PLOS ONE. After careful consideration, we feel that it has merit but does not fully meet PLOS ONE’s publication criteria as it currently stands. Therefore, we invite you to submit a revised version of the manuscript that addresses the points raised during the review process.

The reviewers rightly pointed out the vagueness of the protocol objective. The review authors stated “the main objective of our systematic review and meta-analysis is to determine how does working in the civil construction industry impacts on lung cancer risk for workers” which is not clear what they intend to do. Authors’ three research questions telling exactly the same thing in different ways. They are in fact examining an association between occupational exposure in the civil construction industry and lung cancer. They need to clarify whether they are assessing lung cancer events or lung cancer mortality or both.

Electronic searches. It would be relevant to consider Medline instead of PubMed. Authors are using funnel plot to report publication bias. However, the funnel plot only provides visual impressions. So the author should use Egger's test as well for testing publication bias. Give the version of RevMan that will be used: the latest "5.3"? Discuss the dissemination of findings in the Discussion section.

We look forward to receiving your revised manuscript.

Kind regards,

Rakibul M Islam

Academic Editor

PLOS ONE

Journal Requirements:

Reviewers' comments:

Reviewer's Responses to Questions

**Comments to the Author**

1. Does the manuscript provide a valid rationale for the proposed study, with clearly identified and justified research questions?

Reviewer #1: No

Reviewer #2: Yes

2. Is the protocol technically sound and planned in a manner that will lead to a meaningful outcome and allow testing the stated hypotheses?

Reviewer #1: Partly

Reviewer #2: Yes

3. Is the methodology feasible and described in sufficient detail to allow the work to be replicable?

Reviewer #1: Yes

Reviewer #2: Yes

4. Have the authors described where all data underlying the findings will be made available when the study is complete?

Reviewer #1: Yes

Reviewer #2: Yes

5. Is the manuscript presented in an intelligible fashion and written in standard English?

Reviewer #1: Yes

Reviewer #2: No

6. Review Comments to the Author

You may also provide optional suggestions and comments to authors that they might find helpful in planning their study.

Reviewer #1: Manuscript title: Lung cancer risk in workers of the civil construction industry: A protocol for a systematic review and meta-analysis

Journal Title: PLOS ONE

A) Title

The Authors tried to address the Lung cancer risk in workers of the civil construction industry. Later, the will pool the incidence of type of lung cancer among these populations and the title should be modified in this context.

Suggestion: “Incidence of lung cancer and mortality among civil construction industry workers: A protocol for a systematic review and meta-analysis”

B) Background

The authors didn’t show the problem associated with exposure to carcinogenic substances. What are these chemicals? What have been done so far? What are the gaps identified? What this systematic review going to investigate? How this review is different from the previous studies?

The findings of previous observational studies were consistent, but the authors considered the justification as there is no systematic review on this title. Do you think absence of systematic review must be the justification to carry out systematic review? To my understanding, systematic review should be conducted if:

- The existing studies are contradicting

- Existing studies recommending

- New studies are coming out

I have tried to see some literatures on incidence of /exposure risk as you mentioned/ lung cancer associated with chemical exposure among industrial workers, but there are a couple of reviews and systematic reviews on specific chemical exposure and risk of cancer among these populations. For example; Alonso-Sandra et al: Association between Occupational Exposure to Wood Dust and Cancer: A Systematic Review and Meta-Analysis; Danato et al: Mortality and cancer morbidity among cement production workers: a meta analysis; Boschman et al: Occupational demands and health effects for bricklayers and construction supervisors: A systematic review and some others reporting the incidence of lung cancer as well.

C) Method and material

- I would like to appreciate the authors for their tremendous work on method. It was coherent, detailed and interesting methodology but I have some minor comments: PICOs seems randomized trials. You are going to include studies not participants and you don’t need to mention gender but studies conducted on certain age, gender, etc..

- Why don’t you include prospective Cohort, comparative cross-sectional and randomized trials

- Your pan of data analysis will be with review manager, but you plan to perform regression analysis, how come? Review manager can’t perform regression but R and STATA and other software can.

- Is publication bias not your issue??

- What about your pan on overall quality of evidence? You don’t use the GRADEpro?

D) Discussion

- The plan of discussion is very narrow

- How will you compare your systematic review with the existing reviews?

- What do you think will be the political and health care implication of your study?

Reviewer #2: Dear Editor,

Thank you for providing me the opportunity to review this protocol. The protocol is well devised and covers most aspects needed to answer the study question. However, I have some comments listed below:

Introduction

• Line 45 and 49: The authors should include more recent references for the statistics if available.

• Line 67: Could the authors be clear in presenting if the finding is of a primary study or a review paper?

Review Objectives

• Line 88: The protocol does not appear to state the main objective precisely. Is the object to determine "how does it impact" or "quantification of the risk if analyzed". The language of the objective might create confusion.

Methods/design

• Line 101: The authors should revise the PROSPERO protocol to match with the submitted manuscript. Please provide the dates of the planned search in the abstract at the end of line 23 as from date 'MM/YYYY' to date 'MM/YYYY' (is it until November 19, 2019?) If so, the authors should be able to provide an adapted PubMed search strategy with the number of studies retrieved alongside as an appendix.

Participants/Population

• Line 123: Age restriction is excluded in the manuscript which is inconsistent with the PROSPERO protocol. Please ensure uniformity.

Search Strategy

• Line 150: I would recommend that authors also search Embase database (as mentioned in PROSPERO protocol). Else, the search might result limited studies.

Study selection

• Line 170: Mention the reference manager "XXX" at the first use rather than below.

• Line 179: Use name "PRISMA flow diagram" rather than flow chart.

Other comments

• How will the authors evaluate the quality of evidence? Do they plan to use any tool?

• What are the criteria under which data will be synthesized? (e.g; minimum number of studies to pool the results) In line 212, authors state "extremely high heterogeneity" which is unclear. What is the level of consistency required for synthesis in this review (< 75% or <85% or something else?)

• The authors should include a sub heading Dissemination or combine Ethics and Dissemination.

Optional suggestions to authors

The manuscript appears to be comprehensive, but the authors need to improve some common errors in language which hampers the readability of the paper (e.g; use of full forms more than once, minor spelling errors).

7. PLOS authors have the option to publish the peer review history of their article (what does this mean?). If published, this will include your full peer review and any attached files.

Reviewer #1: No

Reviewer #2: **Yes: **Pratik Pokharel

---

## [Author Response · Author response to Decision Letter 0]

30 Jan 2021

Dear Prof. Rakibul,

We shall be very much appreciated if you may consider the revised version of our paper entitled (former title) " Lung cancer risk in workers of the civil construction industry: A protocol for a systematic review and meta-analysis” by Schnorr and colleagues, suitable for publication in PLOS ONE. Please find below point-by-point responses to the referee’s comments. 

We greatly appreciated the editor´s and reviewer’s comments and suggestions. In this version, we performed the revision of our manuscript according to all their requests (please, kindly find the letter below). We do hope that all the present corrections could attend the editor and reviewer’s requirements for publication in PLOS ONE.

With best regards.

Yours sincerely,

Carlos Eduardo Schnorr (on behalf of all authors)

---

## [Decision Letter · Decision Letter 1]

4 Mar 2021

PONE-D-20-29581R1

Incidence of lung cancer and mortality among civil construction industry workers: A protocol for a systematic review and meta-analysis

PLOS ONE

Dear Dr. Schnorr

Thank you for submitting revised version of your manuscript to PLOS ONE. The manuscript has been improved substantially after addressing reviewers' comments. However, we invite you to submit the manuscript with minor revision that addresses the points raised during the review process.

We look forward to receiving your revised manuscript.

Kind regards,

Rakibul M. Islam, MPH, PhD

Academic Editor

PLOS ONE

Journal Requirements:

Reviewers' comments:

Reviewer's Responses to Questions

**Comments to the Author**

1. Does the manuscript provide a valid rationale for the proposed study, with clearly identified and justified research questions?

Reviewer #1: Yes

Reviewer #2: Yes

2. Is the protocol technically sound and planned in a manner that will lead to a meaningful outcome and allow testing the stated hypotheses?

Reviewer #1: Yes

Reviewer #2: Yes

3. Is the methodology feasible and described in sufficient detail to allow the work to be replicable?

Reviewer #1: Yes

Reviewer #2: Yes

4. Have the authors described where all data underlying the findings will be made available when the study is complete?

Reviewer #1: Yes

Reviewer #2: Yes

5. Is the manuscript presented in an intelligible fashion and written in standard English?

Reviewer #1: Yes

Reviewer #2: No

6. Review Comments to the Author

You may also provide optional suggestions and comments to authors that they might find helpful in planning their study.

Reviewer #1: I would appreciate the authors for their tremendous work and rigorous response on all concerns being forwarded. I have tried to see the manuscript thoroughly and it was well improved substantially.

finally, I would recommend publication in PLOS ONE

Reviewer #2: The manuscript is well revised to improve the transparency, clarity and reproducibility of the planned work. The authors have addressed the comments and the responses are clear. However, I still have some minor comments listed below.

Ethics

• I would recommend modifying the tone of Ethics statement to make it sound better. It can be toned as "This review does not require ethical approval as the review is entirely based on the published data of the ethically approved primary studies."

Study selection

• Line 190: "where agreement cannot be reached” is dispensable.

Discussion

• Improvement on discussion section is fitting. However, it feels as the authors have too many introductory statements on lung cancer evidence which is not relevant here (Line 287-295). How does the evidence provided here link up with the advantages of the paper presented thereafter as "In this way, the results obtained from our study will..."? I would recommend summarising it. Introduction section should be better for presenting the facts and statistics on the grave nature of the outcome due to the risks among construction industry workers.

• The dissemination part is well addressed and can be a separate paragraph.

7. PLOS authors have the option to publish the peer review history of their article (what does this mean?). If published, this will include your full peer review and any attached files.

Reviewer #1: **Yes: **Semagn Mekonnen Abate

Reviewer #2: **Yes: **Pratik Pokharel

---

## [Author Response · Author response to Decision Letter 1]

15 Mar 2021

EDITORS' COMMENTS

The authors: We have reviewed the reference list text to make sure that it is complete and correct. We doble checked each reference from the list in PubMed and Retraction Watch Database (http://retractiondatabase.org/) and found no retracted papers in our reference list. However, we found and included one erratum for the reference “Consonni D, De Matteis S, Pesatori AC, Cattaneo A, Cavallo DM, Lubin JH, et al. Increased lung cancer risk among bricklayers in an Italian population-based case-control study. Am J Ind Med. 2012 May;55(5):423–8.” Both the original and the erratum are now cited in the manuscript. We also revised the reference for the Cochrane Handbook for Systematic Reviews of Interventions. We now correctly cite the handbook as “Higgins JP, Green S, editors. Cochrane Handbook for Systematic Reviews of Interventions Version 5.1.0 [updated March 2011]. 2011. 639 p.” Finally, we found one missing reference in the manuscript (line 288) and the reference is now cited as “British Health and Safety Executive. Occupational Cancer statistics in Great Britain, 2020. London; 2020.”

REVIEWERS' COMMENTS

REVIEWER #1: 

I would appreciate the authors for their tremendous work and rigorous response on all concerns being forwarded. I have tried to see the manuscript thoroughly and it was well improved substantially. Finally, I would recommend publication in PLOS ONE.

The authors: Thank you for all your previous observations and questions. They have helped us to improve the manuscript of the proposed systematic review and meta-analysis.

REVIEWER #2:

The manuscript is well revised to improve the transparency, clarity and reproducibility of the planned work. The authors have addressed the comments and the responses are clear. However, I still have some minor comments listed below:

A) Ethics

• I would recommend modifying the tone of Ethics statement to make it sound better. It can be toned as "This review does not require ethical approval as the review is entirely based on the published data of the ethically approved primary studies."

The authors: We very much appreciated the reviewer´s suggestion, and the Ethics statement has been modified to sound better. The Ethics statement changed from “The present study does not require ethical approval since the original studies must have been conducting according to each local ethics committee.” to “This review does not require ethical approval as the review is entirely based on the published data of the ethically approved primary studies.”

B) Study selection

• Line 190: "where agreement cannot be reached” is dispensable.

The authors: This has been corrected accordingly.

C) Discussion

• Improvement on discussion section is fitting. However, it feels as the authors have too many introductory statements on lung cancer evidence which is not relevant here (Line 287-295). How does the evidence provided here link up with the advantages of the paper presented thereafter as "In this way, the results obtained from our study will..."? I would recommend summarising it. Introduction section should be better for presenting the facts and statistics on the grave nature of the outcome due to the risks among construction industry workers.

The authors: Thank you for all your observations and suggestions. We concentrated the facts and statistics in the introduction section. And we have summarized the statements on lung cancer evidence in the Discussion section, to improve the section clarity and content.

• The dissemination part is well addressed and can be a separate paragraph.

The authors: This has been revised accordingly.

---

## [Editor Report · Decision Letter 2]

6 Apr 2021

Incidence of lung cancer and mortality among civil construction industry workers: A protocol for a systematic review and meta-analysis

PONE-D-20-29581R2

Dear Dr. Schnorr, 

We’re pleased to inform you that your manuscript has been judged scientifically suitable for publication and will be formally accepted for publication once it meets all outstanding technical requirements.

Kind regards,

Rakibul M. Islam, MPH, PhD

Academic Editor

PLOS ONE

---

## [Editor Report · Acceptance letter]

14 Apr 2021

PONE-D-20-29581R2 

Incidence of lung cancer and mortality among civil construction industry workers: A protocol for a systematic review and meta-analysis 

Dear Dr. Schnorr:

I'm pleased to inform you that your manuscript has been deemed suitable for publication in PLOS ONE. Congratulations! Your manuscript is now with our production department. 

Kind regards, 

on behalf of

Dr. Rakibul M. Islam 

Academic Editor

PLOS ONE